# FUZZY c-MEANS CLUSTERING FOR PERSISTENCE DIAGRAMS

## ABSTRACT

Persistence diagrams concisely represent the topology of a point cloud whilst having strong theoretical guarantees. Most current approaches to integrating topological information into machine learning implicitly map persistence diagrams to a Hilbert space, resulting in deformation of the underlying metric structure whilst also generally requiring prior knowledge about the true topology of the space. In this paper we give an algorithm for Fuzzy c-Means (FCM) clustering directly on the space of persistence diagrams,22 enabling unsupervised learning that automatically captures the topological structure of data, with no prior knowledge or additional processing of persistence diagrams. We prove the same convergence guarantees as traditional FCM clustering: every convergent subsequence of iterates tends to a local minimum or saddle point. We end by presenting experiments where our fuzzy topological clustering algorithm allows for unsupervised top-$k$ candidate selection in settings where (i) the properties of persistence diagrams make them the natural choice over geometric equivalents, and (ii) the probabilistic membership values let us rank candidates in settings where verifying candidate suitability is expensive: lattice structure classification in materials science and pre-trained model selection in machine learning.

## 1 INTRODUCTION

Persistence diagrams, a concise representation of the topology of a point cloud with strong theoretical guarantees, have emerged as a new tool in the field of data analysis (Edelsbrunner & Harer, 2010). Persistence diagrams have been successfully used to analyse problems ranging from financial crashes (Gidea & Katz, 2018) to protein binding (Kovacev-Nikolic et al., 2014), but the non-Hilbertian nature of the space of persistence diagrams means it is difficult to directly use persistence diagrams for machine learning. In order to better integrate diagrams into machine learning workflows, efforts have been made to map them into a more manageable form; primarily through embeddings into finite feature vectors, functional summaries, or by defining a positive-definite kernel on diagram space. In all cases, this explicitly or implicitly embeds diagrams into a Hilbert space which deforms the metric structure, potentially losing important information. With the exception of Topological Autoencoders, techniques to integrate these persistence-based summaries as topological regularisers and loss functions currently require prior knowledge about the correct topology of the dataset, which is clearly not feasible in most scenarios.

Against this background, we give an algorithm to perform Fuzzy c-Means (FCM) clustering (Bezdek, 1980) *directly* on collections of persistence diagrams, giving an important unsupervised learning algorithm and enabling learning from persistence diagrams without deforming the metric structure. We perform the convergence analysis for our algorithm, giving the same guarantees as traditional FCM clustering: that every convergent subsequence of iterates tends to a local minimum or saddle point. We demonstrate the value of our fuzzy clustering algorithm by using it to cluster datasets that benefit from both the topological and fuzzy nature of our algorithm. We apply our technique in two settings: lattice structures in materials science and the decision boundaries of CNNs. A key property for machine learning in materials science has been identified as "invariance to the basis symmetries of physics [...] rotation, reflection, translation" (Schmidt et al., 2019). Geometric clustering algorithms do not have this invariance, but persistence diagrams do, making them ideally suited for this application; we can cluster transformed lattice structure datasets where geometric equivalents fail. In addition to this, our probabilistic membership values allow us to rank the top-$k$

most likely lattices assigned to a cluster. This is particularly important in materials science, as further investigation requires expensive laboratory time and expertise. Our second application is inspired by Ramamurthy et al. (2019), who show that models perform better on tasks if they have topologically similar decision boundaries. We use our algorithm to cluster models and tasks by the persistence diagrams of their decision boundaries. Not only is our algorithm able to successfully cluster models to the correct task, based just on the topology of its decision boundary, but we show that higher membership values imply better performance on unseen tasks.

## 1.1 RELATED WORK

**Means of persistence diagrams.** Our work relies on the existence of statistics in the space of persistence diagrams. Mileyko et al. (2011) first showed that means and expectations are well-defined in the space of persistence diagrams. Specifically, they showed that the Fréchet mean, an extension of means onto metric spaces, is well-defined under weak assumptions on the space of persistence diagrams. Turner et al. (2012) then developed an algorithm to compute the Fréchet mean. We adapt the algorithm by Turner et al. to the weighted case, but the combinatoric nature of their algorithm makes it computationally intense. Lacombe et al. (2018) framed the computation of means and barycentres in the space of persistence diagram as an optimal transport problem, allowing them to use the Sinkhorn algorithm (Cuturi & Doucet, 2014) for fast computation of approximate solutions. The vectorisation of the diagram required by the algorithm by Lacombe et al. makes it unsuitable for integration into our work, as we remain in the space of persistence diagrams. Techniques to speed up the matching problem fundamental to our computation have also been proposed by Vidal et al. (2020) and Kerber et al. (2017).

**Learning with persistence-based summaries.** Integrating diagrams into machine learning work-flows remained challenging even with well-defined means, as the space is non-Hilbertian (Turner & Spreemann, 2019). As such, efforts have been made to map diagrams into a Hilbert space; primarily either by embedding into finite feature vectors (Kališnik, 2018; Fabio & Ferri, 2015; Chepushtanova et al., 2015) or functional summaries (Bubenik, 2015; Rieck et al., 2019), or by defining a positive-definite kernel on diagram space (Reininghaus et al., 2015; Carrière et al., 2017; Le & Yamada, 2018). These vectorisations have been integrated into deep learning either by learning parameters for the embedding (Hofer et al., 2017; Carrière et al., 2020; Kim et al., 2020; Zhao & Wang, 2019; Zieliński et al., 2019), or as part of a topological loss or regulariser (Chen et al., 2018; Gabrielsson et al., 2020; Clough et al., 2020; Moor et al., 2019). However, the embeddings used in these techniques deform the metric structure of persistence diagram space (Bubenik & Wagner, 2019; Wagner, 2019; Carrière & Bauer, 2019), potentially leading to the loss of important information. Furthermore, these techniques generally require prior knowledge of a 'correct' target topology which cannot plausibly be known in most scenarios. In comparison, our algorithm acts in the space of persistence diagrams so it does not deform the structure of diagram space via embeddings, and is entirely unsupervised, requiring no prior knowledge about the topology.

**Hard clustering.** Maroulas et al. (2017) gave an algorithm for hard clustering persistence diagrams based on the algorithm by Turner et al. Lacombe et al. (2018) gave an alternate implementation of hard clustering based on their algorithm for barycentre computation, providing a computational speed-up over previous the work by Maroulas et al. The primary advantages of our work over previous work on hard clustering are as follows.

(i) The probabilistic membership values allow us to rank datasets in the cluster, enabling top-$k$ candidate selection in settings where verifying correctness is expensive. The value provided by this fuzzy information is demonstrated in the experiments.

(ii) The fuzzy membership values provide information about proximity to all clusters, whereas hard labelling loses most of that information. In our experiments we demonstrate that this additional information can be utilised in practice.

(iii) The weighted cost function makes the convergence analysis (which we provide) entirely non-trivial in comparison to the non-fuzzy case. We consider this convergence analysis a primary contribution of our paper.

(iv) Fuzzy membership values have been shown to be more robust to noise than discrete labels (Klawonn, 2004).

(v) Unlike hard clustering, fuzzy clustering is analytically differentiable, allowing integration of the fuzzy clustering step into deep learning methods (Wilder et al., 2019).

**Geometric equivalents.** The most similar unsupervised learning technique to our algorithm is Wasserstein Barycentre Clustering (WBC). It clusters datasets of point clouds by the Wasserstein distance between the point clouds, rather than the Wasserstein distance between their persistence diagrams. We compare our algorithm experimentally to WBC using ADMM (Ye & Li, 2014), Bregman ADMM (Ye et al., 2017), Subgradient Descent (Cuturi & Doucet, 2014), Iterative Bregman Projection (Benamou et al., 2015), and full linear programming (Li & Wang, 2008). Each of these algorithms computes or approximates the Wasserstein barycentre in different ways. Theoretically, fuzzy discrete distribution clustering (d. A. T. de Carvalho et al., 2015) is similar to our algorithm, but the addition of the diagonal in the persistence diagram makes our work distinct.

## 1.2 Our contributions

1. Our main contribution is an algorithm for Fuzzy c-Means clustering of persistence diagrams, along with the convergence analysis. Given a collection of persistence diagrams $\mathbb{D}_1, \ldots, \mathbb{D}_n$, we alternatively calculate cluster centres $\mathbb{M}_1, \ldots, \mathbb{M}_c$ and membership values $r_{jk} \in [0, 1]$ which denote the degree to which diagram $\mathbb{D}_j$ is associated with cluster $\mathbb{M}_k$. We prove Theorem 1, showing that every convergent subsequence of these alternative update steps tends to a local minimum or saddle point of the cost function. This is the same convergence guarantee provided by traditional FCM clustering (Bezdek et al., 1987), but requires additional work as the space of persistence diagrams with the Wasserstein distance has far weaker theoretical properties than Euclidean space.

2. Updating the cluster centres requires computing the weighted Fréchet mean. We extend the algorithm given by Turner et al. (2012) to the weighted case, justifying our addition of weights by extending their proof to show that the updated algorithm converges.

3. We implement our algorithm in Python, available in the supplementary materials. It works with persistence diagrams from commonly used open-source libraries for Topological Data Analysis (TDA),[1] so is available for easy integration into current workflows, offering a powerful unsupervised learning algorithm to data science practitioners using TDA.

4. We demonstrate the application of our algorithm to settings where (i) the properties of persistence diagrams makes clustering them the natural choice over geometric equivalents and (ii) the probabilistic membership values can be used to rank candidates for top-$k$ selection. Our algorithm classifies transformed lattice structures from materials science where geometric equivalents fail, whilst giving probabilistic rankings to help prioritise expensive further investigation. We also cluster the persistence diagrams of decision boundaries and labelled datasets, showing that our fuzzy clustering captures information about model performance on unseen tasks.

## 2 Topological preliminaries

Topological Data Analysis emerged from the study of algebraic topology, providing a toolkit to fully describe the topology of a dataset. We offer a quick summary below; for more comprehensive details see Edelsbrunner & Harer (2010). A set of points in $\mathbb{R}^d$ are indicative of the shape of the distribution they are sampled from. By connecting points that are pairwise within $\epsilon > 0$ distance of each other, we can create an approximation of the distribution called the Vietoris-Rips complex (Vietoris, 1927). Specifically, we add the convex hull of any collection of points that are pairwise at most $\epsilon$ apart to the $\epsilon$-Vietoris-Rips complex. However, choosing an $\epsilon$ remains problematic; too low a value and key points can remain disconnected, too high a value and the points become fully connected. To overcome this we use *persistence*: we consider the approximation over all values of $\epsilon$ simultaneously, and study how the topology of that approximation evolves as $\epsilon$ grows large. We call the collection of complexes for all $\epsilon$ a filtration.

For each $\epsilon$, we compute the $p$-homology group. This tells us the topology of the $\epsilon$-Vietoris-Rips complex: the 0-homology counts the number of connected components, the 1-homology counts the number of holes, the 2-homology counts the number of voids, etc. (Edelsbrunner et al., 2000).

---

[1] Dionysus and Ripser (Bauer, 2019).

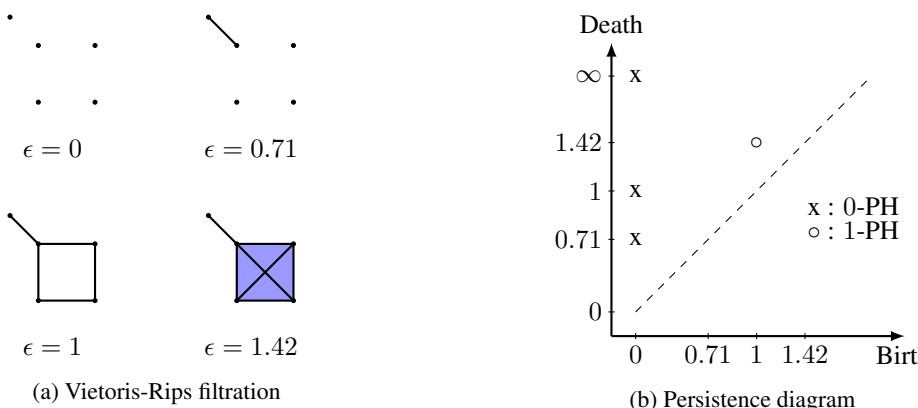

$\epsilon = 0$ $\qquad$ $\epsilon = 0.71$

$\epsilon = 1$ $\qquad$ $\epsilon = 1.42$

(a) Vietoris-Rips filtration

(b) Persistence diagram

Figure 1: An example Vietoris-Rips filtration with its persistence diagram. We only add convex hulls when points are pairwise $\epsilon$ apart, so there is a hole at $\epsilon = 1$ whilst the diagonals of the square are not close enough to fill the hole in. This hole can be seen as a point at $(1, 1.42)$ in the 1-PH persistence diagram.

The $p$-persistent homology ($p$-PH) group is created by summing the $p$-homology groups over all $\epsilon$. This results in a $p$-PH group that summarises information about the topology of the dataset at all granularities. If a topological feature, such as a connected component or hole, persists throughout a large range of granularities, then it's more likely to be a feature of the distribution. If it only persists for a short amount of time, then it's more likely to be noise (Cohen-Steiner et al., 2007). We can stably map a $p$-PH group into a multiset in the extended plane called a persistence diagram (Chazal et al., 2012). Each topological feature has a birth and death: a feature is born when it enters the complex, and dies when the complex grows enough to destroy it. For example, in Figure 1(a), a feature is born at $\epsilon = 1$ when four lines form a hole. This feature dies at $\epsilon = 1.42$ when the hole is filled in. This is shown in the persistence diagram in Figure 1(b) as a point at $(1, 1.42)$ in 1-PH. By computing the birth/death points for each topological feature in the filtration, we get a complete picture of the topology of the point cloud at all granularities (Zomorodian & Carlsson, 2005). The persistence diagram is the collection of birth/death points, along with the diagonal $\Delta = \{(a, a) : a \in \mathbb{R}\}$ with infinite multiplicity, added in order to make the space of persistence diagrams complete (Mileyko et al., 2011).

## 3 ALGORITHMIC DESIGN

### 3.1 CLUSTERING PERSISTENCE DIAGRAMS

In order to cluster we need a distance on the space of persistence diagrams. We use the 2-Wasserstein $L_2$ metric as it is stable on persistence diagrams of finite point clouds (Chazal et al., 2012). The Wasserstein distance is an optimal transport metric that has found applications across machine learning. In the Euclidean case, it quantifies the smallest distance between optimally matched points. Given diagrams $\mathbb{D}_1, \mathbb{D}_2$, the distance is

$$W_2(\mathbb{D}_1, \mathbb{D}_2) = \left( \inf_{\gamma : \mathbb{D}_1 \to \mathbb{D}_2} \sum_{x \in \mathbb{D}_1} ||x - \gamma(x)||_2^2 \right)^{1/2},$$

where the infimum is taken over all bijections $\gamma : \mathbb{D}_1 \to \mathbb{D}_2$. Note that as we added the diagonal with infinite multiplicity to each diagram, these bijections exist. If an off-diagonal point is matched to the diagonal the transportation cost is simply the shortest distance to the diagonal. In fact, the closer a point is to the diagonal, the more likely it is to be noise (Cohen-Steiner et al., 2007), so this ensures our distance is not overly affected by noise.

We work in the space $\mathcal{D}_{L^2} = \{\mathbb{D} : W_2(\mathbb{D}, \Delta) < \infty\}$,[2] as this leads to a geodesic space with known structure (Turner et al., 2012). Given a collection of persistence diagrams $\{\mathbb{D}_j\}_{j=1}^n \subset \mathcal{D}_{L^2}$ and a fixed number of clusters $c$, we wish to find cluster centres $\{\mathbb{M}_k\}_{k=1}^c \subset \mathcal{D}_{L^2}$, along with membership values $r_{jk} \in [0,1]$ that denote the extent to which $\mathbb{D}_j$ is associated with cluster $\mathbb{M}_k$. We follow probabilistic fuzzy clustering, so that $\sum_k r_{jk} = 1$ for each $j$.

We extend the FCM algorithm originally proposed by Bezdek (1980). Our $r_{jk}$ is the same as traditional FCM clustering, adapted with the Wasserstein distance. That is,

$$r_{jk} = \left( \sum_{l=1}^c \frac{W_2(\mathbb{M}_k, \mathbb{D}_j)}{W_2(\mathbb{M}_l, \mathbb{D}_j)} \right)^{-1}. \tag{1}$$

To update $\mathbb{M}_k$, we compute the weighted Fréchet mean $\hat{\mathbb{D}}$ of the persistence diagrams $\{\mathbb{D}_j\}_{j=1}^n$ with the weights $\{r_{jk}^2\}_{j=1}^n$. Specifically,

$$\mathbb{M}_k \longleftarrow \arg\min_{\hat{\mathbb{D}}} \sum_{j=1}^n r_{jk}^2 W_2(\hat{\mathbb{D}}, \mathbb{D}_j)^2, \text{ for } k = 1, \ldots, c. \tag{2}$$

As the weighted Fréchet mean extends weighted centroids to general metric spaces, this gives our fuzzy cluster centres. The computation of the weighted Fréchet mean is covered in Section 3.2. By alternatively updating (1) and (2) we get a sequence of iterates. Theorem 1, proven in Appendix A, provides the same convergence guarantees as traditional FCM clustering.

**Theorem 1.** *Every convergent subsequence of the sequence of iterates obtained by alternatively updating membership values and cluster centres with (1) and (2) tends to a local minimum or saddle point of the cost function* $J(R, \mathbb{M}) = \sum_{j=1}^n \sum_{k=1}^c r_{jk}^2 W_2(\mathbb{M}_k, \mathbb{D}_j)^2$.

Observe that we only guarantee the convergence of subsequences of iterates. This is the same as traditional FCM clustering, so we follow the same approach to a stopping condition and run our algorithm for a fixed number of iterations. The entire algorithm is displayed in Algorithm 1.

---

**Algorithm 1** FPDCluster
---

    **Input** Diagrams $\mathbb{D} = \{\mathbb{D}_j\}_{j=1}^n$, number of clusters $c$, maximum iterations MAXITER
    **Output** Cluster centres $\mathbb{M} = \{\mathbb{M}_k\}_{k=1}^c$, membership values $R = \{r_{jk}\}$

  1: $\mathbb{D} = \text{ADDDIAGONALS}(\mathbb{D})$
  2: $\mathbb{M} = \text{INITCENTRES}(\mathbb{D})$
  3: **for** count in 1..MAXITER **do**
  4:     **for** $j$ in 1..$n$ **do**
  5:         **for** $k$ in 1..$c$ **do**
  6:             $r_{jk} \leftarrow \left( \sum_{l=1}^c \frac{W_2(\mathbb{M}_k, \mathbb{D}_j)}{W_2(\mathbb{M}_l, \mathbb{D}_j)} \right)^{-1}$
  7:         **end for**
  8:     **end for**
  9:     **for** $k$ in 1..$c$ **do**
10:         $\mathbb{M}_k \leftarrow \text{WFRECHETMEAN}(\mathbb{D}, R_k)$
11:     **end for**
12: **end for**
13: **return** $\mathbb{M}, R$

---

### 3.2 COMPUTING THE WEIGHTED FRÉCHET MEAN

Turner et al. (2012) give an algorithm for the computation of Fréchet means. In this section we extend their algorithm and proof of convergence to the weighted case. In Algorithm 1 we add copies of the diagonal to ensure that each diagram has the same cardinality; denote this cardinality as $m$. To compute the weighted Fréchet mean, we need to find $\mathbb{M}_k = \{y^{(i)}\}_{i=1}^m$ that minimises the Fréchet function in (2). Implicit to the Wasserstein distance is a bijection $\gamma_j : y^{(i)} \mapsto x_j^{(i)}$ for

---

[2]To ensure that our persistence diagrams are all in this space, we map points at infinity to a hyperparameter $T$ that is much larger than other death values in the diagram. This hyperparameter ensures that the one point at infinity will always be matched to the corresponding point at infinity when computing the Wasserstein distance between diagrams. This can be avoided entirely by computing the diagrams with extended persistence (Cohen-Steiner et al., 2009), which removes points at infinity.

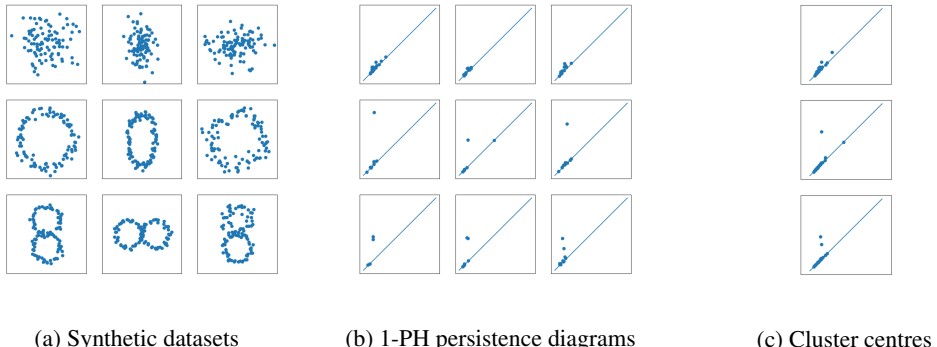

(a) Synthetic datasets      (b) 1-PH persistence diagrams      (c) Cluster centres

Figure 2: Our algorithm successfully clustered the persistence diagrams, finding cluster centres that represent the topology of the datasets. The cluster centres have zero, one, or two significant off-diagonal points, representing zero, one, or two holes in the datasets.

each $j$. Supposing we know these bijections, we can rearrange the Fréchet function into the form $F(\mathbb{M}_k) = \sum_{j=1}^{n} r_{jk}^2 W_2(\mathbb{M}_k, \mathbb{D}_j)^2 = \sum_{i=1}^{m} \sum_{j=1}^{n} r_{jk}^2 ||y^{(i)} - x_j^{(i)}||^2$.

In this form, the summand is minimised for $y^{(i)}$ by the weighted Euclidean centroid of the points $\{x_j^{(i)}\}_{j=1}^{n}$. Therefore to compute the weighted Fréchet mean, we need to find the correct bijections. We start by using the Hungarian algorithm to find an optimal matching between $\mathbb{M}_k$ and each $\mathbb{D}_j$. Given a $\mathbb{D}_j$, for each point $y^{(i)} \in \mathbb{M}_k$, the Hungarian algorithm will assign an optimally matched point $x_j^{(i)} \in \mathbb{D}_j$. Specifically, we find matched points

$$\left[x_j^{(i)}\right]_{i=1}^{m} \longleftarrow \text{Hungarian}(\mathbb{M}_k, \mathbb{D}_j), \text{ for each } j = 1, \dots, n. \tag{3}$$

Now, for each $y^{(i)} \in \mathbb{M}_k$ we need to find the weighted average of the matched points $\left[x_j^{(i)}\right]_{j=1}^{n}$. However, some of these points could be copies of the diagonal, so we need to consider three distinct cases: that each matched point is off-diagonal, that each one is a copy of the diagonal, or that the points are a mixture of both. We start by partitioning $1, \dots, n$ into the indices of the off-diagonal points $\mathcal{J}_{\text{OD}}^{(i)} = \left\{j : x_j^{(i)} \neq \Delta\right\}$ and the indices of the diagonal points $\mathcal{J}_{\text{D}}^{(i)} = \left\{j : x_j^{(i)} = \Delta\right\}$ for each $i = 1, \dots, m$. Now, if $\mathcal{J}_{\text{OD}} = \emptyset$ then $y^{(i)}$ is a copy of the diagonal. If not, let $w = \left(\sum_{j \in \mathcal{J}_{\text{OD}}^{(i)}} r_{jk}^2\right)^{-1} \sum_{j \in \mathcal{J}_{\text{OD}}^{(i)}} r_{jk}^2 x_j^{(i)}$ be the weighted mean of the off-diagonal points. If $\mathcal{J}_{\text{D}}^{(i)} = \emptyset$, then $y^{(i)} = w$. Otherwise, let $w_\Delta$ be the point on the diagonal closest to $w$. Then our update is

$$y^{(i)} \longleftarrow \frac{\sum_{j \in \mathcal{J}_{\text{OD}}^{(i)}} r_{jk}^2 x_j^{(i)} + \sum_{j \in \mathcal{J}_{\text{D}}^{(i)}} r_{jk}^2 w_\Delta}{\sum_{j=1}^{n} r_{jk}^2}, \text{ for } i = 1, \dots, m. \tag{4}$$

We alternate between (3) and (4) until the matching remains the same. Theorem 2, proving that this algorithm converges to a local minimum of the Fréchet function, is proven in Appendix B. Also in Appendix B is Algorithm 2, giving an overview of the entire computation.

**Theorem 2.** *Given diagrams $\mathbb{D}_j$, membership values $r_{jk}$, and the Fréchet function $F(\hat{\mathbb{D}}) = \sum_{j=1}^{n} r_{jk}^2 W_2(\hat{\mathbb{D}}, \mathbb{D}_j)^2$, then $\mathbb{M}_k = \{y^{(i)}\}_{i=1}^{m}$ is a local minimum of $F$ if and only if there is a unique optimal pairing from $\mathbb{M}_k$ to each of the $\mathbb{D}_j$ and each $y^{(i)}$ is updated via (4).*

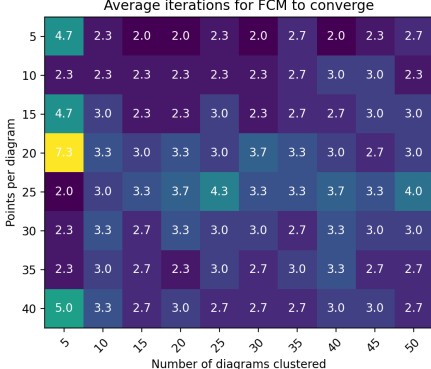 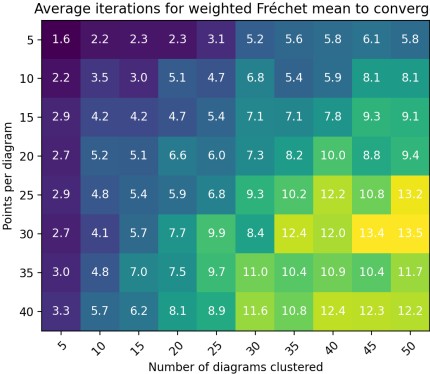

Figure 3: Heatmaps showing average number of iterations for fuzzy clustering of persistence diagrams (left) and the weighted Fréchet mean computation (right) to converge. Convergence of the FCM algorithm is determined when the cost function is stable to within $\pm 0.5\%$. Convergence experiments were carried out on randomly generated persistence diagrams with three repeats.

## 4 EXPERIMENTS

### 4.1 SYNTHETIC DATA

**Exemplar clustering.** We start by demonstrating our algorithm on a simple synthetic dataset designed to highlight its ability to cluster based on the topology of the underlying datasets. We produce three datasets of noise, three datasets of a ring, and three datasets of figure-of-eights, all shown in Figure 2(a). In Figure 2(b) we show the corresponding 1-PH persistence diagrams. Note that the persistence diagrams have either zero, one, or two significant off-diagonal points, corresponding to zero, one, or two holes in the datasets. We then use our algorithm to cluster the nine persistence diagrams into three clusters. Having only been given the list of diagrams, the number of clusters, and the maximum number of iterations, our algorithm successfully clusters the diagrams based on their topology. Figure 2(c) shows that the cluster centres have zero, one, or two off-diagonal points: our algorithm has found cluster centres that reflect the topological features of the datasets. Because we are reducing the cardinality and dimensionality of datasets by mapping into persistence diagrams, we also demonstrate a speed-up of at least an order of magnitude over Wasserstein barycentre clustering methods. Details of these timing experiments are in Appendix C.1.[3]

**Empirical behaviour.** Figure 4 shows the results of experiments run to determine the empirical performance of our algorithm. We give theoretical guarantees that every convergent subsequence will tend to a local minimum, but in practice it remains important that our algorithm will converge, and within a reasonable timeframe. To this end we ran experiments on a total of 825 randomly generated persistence diagrams, recording the number of iterations and cost functions for both the FCM clustering and the weighted Fréchet mean (WFM) computation. We considered the FCM to have converged when the cost function remained within $\pm 0.5\%$. As explained in Section 3.2, the WFM converges when the matching stays the same. Our experiments showed that the FCM clustering consistently converges within <5 iterations, regardless of the number of diagrams and points per diagram (note that the *time per iteration* increases as the number of points/diagrams increases, even if the number of iterations remains stable). We had no experiments in which the algorithm did not converge. The WFM computation requires more iterations as both number of diagrams and number of points per diagram increases, but we once again experienced no failures to converge in each of our experiments. In general, running the algorithm offered no difficulties on a laptop, and we believe the algorithm is ready for use by practitioners in the TDA community.

---

[3]Further details of all experiments are available in Appendix C.

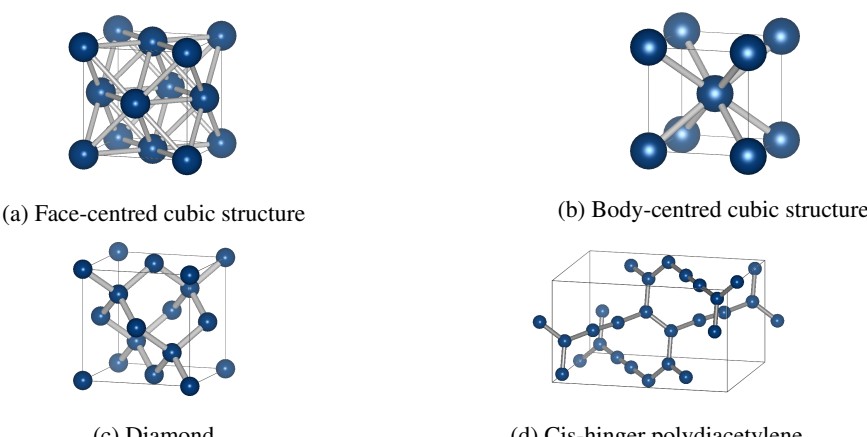

(a) Face-centred cubic structure

(b) Body-centred cubic structure

(c) Diamond

(d) Cis-hinger polydiacetylene

Figure 4: Cubic structures (top) and carbon allotropes (bottom). Our algorithm can cluster transformed lattice structures where comparable geometric algorithms fail.

## 4.2 LATTICE STRUCTURES

A key property for machine learning in materials science has been identified as "invariance to the basis symmetries of physics [...] rotation, reflection, translation" (Schmidt et al., 2019). Removing the need for a standardised coordinate system allows machine learning methods to be applied to a broader range of existing coordinate datasets generated by experimental methods (e.g., x-ray diffraction and scanning electron microscope imaging) and computational methods (e.g., density functional theory). Persistence diagrams, which capture affine transformation-invariant properties of datasets, are ideally suited for application in this domain. Our algorithm enables successful unsupervised learning on these datasets for the first time. Additionally, the fuzzy membership values allow top-$k$ ranking of candidates suggested by our algorithm. This is particularly important in materials science, where further investigation of materials can be extremely costly.

The large majority of solids are comprised of lattices: regularly repeating unit cells of atoms. This lattice structure directly determines the properties of a material (Hull & Bacon, 2011) and it has been predicted that machine learning will reveal presently unknown links between structure and property by identifying new trends across materials (Meredig, 2019; Wei et al., 2019). We apply our algorithm to two examples of lattice structures from materials science: cubic structures and carbon allotropes. Cubic structures are important because they are ubiquitous. The most common lattice structures, particularly amongst pure metals, are face-centred cubic (FCC) structures and body-centred cubic (BCC) structures (Putnis, 1992), shown in Figures 4(a) and 4(b). Carbon allotropes, such as graphene and diamond, are widely anticipated to revolutionise electronics and optoelectronics (Wang et al., 2016). We focus on the carbon allotropes diamond and cis-hinged polydiacetylene, shown in Figures 4(c) and 4(d).

We use atomic positions for the unit-cells of iron mp-150 and iron mp-13 from the Materials Project (Jain et al., 2013), representing BCC and FCC structures respectively, for our first experiment. For our second experiment we use diamond and cis-hinged polydiacetylene unit-cell atomic positions from the Samara Carbon Allotrope Database (Hoffmann et al., 2016). We simulate distinct collections of lattices by transforming the atomic coordinates, with no information about bonds given to the algorithms. The properties of persistence diagrams mean that we can successfully cluster the atomic coordinates derived from the same base unit-cell regardless of the transformations applied to the coordinate system, fulfilling the key property identified above (we consider a clustering successful when all datasets from the same lattice structure have their highest membership values for the same cluster). In comparison, we run Wasserstein barycentre clustering on the same datasets using several state-of-the-art algorithms for barycentre computation and approximation. Each can only successfully cluster the cubic structures after reflection, and none of them successfully cluster the carbon allotropes after any transformations. We give specific values for these results in Appendix C.2.

### 4.3 DECISION BOUNDARIES

Learnt models have been shown to perform better on datasets which have a similar persistence diagram to the model's decision boundary (Ramamurthy et al., 2019). In fact, topological complexity has been shown to correlate with generalisation ability (Guss & Salakhutdinov, 2018; Gabrielsson & Carlsson, 2019; Rieck et al., 2018). We utilise our algorithm to cluster the topology of models and tasks, showing that high membership values imply better performance on tasks. Specifically, given a dataset with $n$ classes, we fix one class to define $n-1$ *tasks*: binary classification of the fixed class vs each of the remaining classes. On each of these tasks, we train a *model*. We compute the decision boundary of the model $f$, defined as $(x_1, \ldots, x_m, f(x))$ where $f(x)$ is the model's prediction for $x = (x_i)_i$, and the decision boundary of the tasks, defined via the labelled dataset as $(x_1, \ldots, x_m, y)$ where $y$ is the true label. We compute the 1-persistence diagrams of the tasks' and models' decision boundaries and cluster them to obtain membership values and cluster centres. To view task and model proximity through our clustering, we find the cluster centre with the highest membership value for each task, and consider the models closest to that cluster centre. Note that in general you cannot do this with hard clustering: most of the time a path will not exist from task to cluster centre to model, because each task/model is only associated with a single cluster. This contrasts with fuzzy clustering, where you have information about how close each model/task is to each cluster centre. We further discuss why this does not work for hard clustering in Appendix C.3.

To assess the ability of our model/task clustering, we performed the above experiment on three different datasets: MNIST (LeCun et al., 2010), FashionMNIST (Xiao et al., 2017), and Kuzushiji-MNIST (Clanuwat et al., 2018). We repeat each experiment three times. Our goal is to evaluate whether or not the clustering is capturing information about model performance on tasks, so as a baseline we use the average performance of all models on a fixed task, averaged over all tasks. Recall that we are just using the topology of the decision boundary: no information about model performance on the task was used until we assessed the ability of the clustering to capture information about the tasks. We start by verifying what happens if we use the model closest to the cluster centre associated with the task (i.e., top-1). We see a significant increase in performance, indicating that the topological fuzzy clustering has selected the model trained on the task: this means the clustering is working. We also compute the top-3 and top-2 performance change over average. We still see a statistically significant increase in performance over average performance, indicating that our membership values are capturing information about model performance on unseen tasks. These results are shown in Table 1. By successfully matching models to tasks using fuzzy clustering, we offer a new technique for future model marketplaces, demonstrating the value of topological clustering when analysing decision boundaries.

Table 1: Performance increase/decrease over average task performance when using learnt fuzzy membership values for model selection. The increase in performance demonstrates that our fuzzy clustering automatically clusters models near tasks they perform well on.

| | Performance change vs random model selection (%) | | |
|---|---|---|---|
| | Top-3 | Top-2 | Top-1 |
| MNIST | $+6.17^{\pm 2.18}$ | $+10.81^{\pm 1.88}$ | $+20.88^{\pm 4.08}$ |
| FashionMNIST | $+16.46^{\pm 4.00}$ | $+21.94^{\pm 4.73}$ | $+23.30^{\pm 8.72}$ |
| Kuzushiji | $+6.61^{\pm 1.78}$ | $+11.18^{\pm 2.45}$ | $+21.89^{\pm 5.54}$ |

## 5 CONCLUSIONS

We have developed FCM clustering on the space of persistence diagrams, adding an important class of unsupervised learning to Topological Data Analysis' toolkit. Our topological fuzzy clustering paves the way for applications in materials science and pre-trained model marketplaces, where we envisage our algorithm being used as part of automated materials discovery and model selection.

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
