# OpenReview forum: "Fuzzy c-Means Clustering for Persistence Diagrams"
_ICLR.cc/2021/Conference — Reject_

### Official Review · AnonReviewer3 · 2020-10-27
**Sound approach with weaknesses in the experiments**

**Rating:** 6
**Confidence:** 4

**Review:**

The authors propose a novel algorithm for the fuzzy clustering of persistence diagrams. To determine cluster centroids, the Wasserstein-2 distance is used to minimize the weighted Fr\’echet mean between a potential cluster center and all PDs considered for clustering. The authors proof convergence of the clustering algorithm and conduct experiments on 1) synthetic data, 2) lattice structures, and 3) decision boundaries of neural networks. In the last experiment is was shown that models whose PDs cluster close to the PDs of a given task lead to higher classification performance than random classifiers, demonstrating the merit of PDs as a useful tool for model selection.

The paper is very well written and clearly presented. Even though I did not check all proofs in the appendix, it seems like a mathematically sound approach that leads to good clustering results. The contributions and the relevance of the topic are apparent; however, I think the experimental section is weaker than it could be. While I like the experiments on the synthetic data set and the decision boundaries, I think the weakest part of the paper are the experiments on the lattice structures. Respective paragraphs are packed with technical details which are not crucial to understand the problem at hand. The results are very qualitative and outsourced to the appendix. Furthermore, Figure 3 is interesting to look at and to get a better understand of the structural differences of the data; however, it would be nice to have visual representations of the results of 4.2. For example, would it be possible to choose $c=4$ and find the four different categories that you show in Figure 3 (by selecting 10-20 structures from each category and run your algorithm on them)? You could visualize a representative of each category (like in the current figure) and their respective PDs (just plotted on top of each other for each category). In a third panel, it would be interesting to see a representation of the centroids and all PDs using Multidimensional Scaling based on the W2 distance. I would also recommend using a more quantitative way of evaluating cluster quality (e.g. [1]).
A second weakness I see in the experiments is that in the introduction, you mention previous work on fuzzy discrete distribution clustering (by de Carvalho et al.). Still, you do not compare your results to their method (does adding the diagonal make such a big difference?).

Minor comments and questions:

•	Related work: When you mention the correlation of topological complexity with generalization ability, you miss the work by Rieck et al. [2].

•	I am a fan of citing software libraries when they provide how to in their website. Hence I would recommend citing Ripser [3] (see https://github.com/Ripser/ripser for BibTeX entry) and potentially other Software you used in the main paper.

•	You say “[…] the vectorization […] required by Lacombe et al.’s algorithm makes it unsuitable for integration into our work”. Could you elaborate on this? Can you not even choose this as a comparison partner?

•	What do you mean when you say other persistence-based learning strategies required prior knowledge of a ‘correct’ target topology which can’t plausibly be known? The work by Moor et al. for example, extracts persistence directly from the input, which is “trivial” to know since you can compute it.

•	Topological Preliminaries: Why do you introduce the \v{C}ech complex when you use Vietoris-Rips for your experiments?

•	One feat of your approach is that you add tuples on the diagonal to make the cardinality of all compared PDs match. I wonder how susceptible your approach is to increasingly different cardinality (that has to be filled). Maybe you can conduct a small ablation study on your synthetic data set and evaluate how “good” your mean PDs are when cardinalities are increasingly different.

[1]: RJGB Campello, ER Hruschka. A fuzzy extension of the silhouette width criterion for cluster analysis. Fuzzy Sets and Systems, 2006 – Elsevier

[2]: Bastian Rieck, Matteo Togninalli, Christian Bock, Michael Moor, Max Horn, Thomas Gumbsch, Karsten Borgwardt. Neural Persistence: A Complexity Measure for Deep Neural Networks Using Algebraic Topology. ICLR 2019.

[3]: Ulrich Bauer. Ripser: efficient computation of Vietoris-Rips persistence barcodes. Preprint.

======================
Update: Thank you for your rebuttal, I still think this is a promising paper and lean towards acceptance. However, after the discussion and the update of the manuscript, my score will remain the same.

---

> ### Author Response · Authors · 2020-11-16
> **Initial reviewer response**
>
> Hi, thank you for your useful review. We appreciate your kind comments on the 'contribution and relevance of the topics'. We are working to run more experiments based on your (and other reviewers) feedback. Your comments on how to specifically improve the experiments section and our figures are particularly useful, thank you. We'll upload a new version of the paper before the end of the rebuttal period. We respond to specific comments below.
>
> 'You mention previous work on fuzzy discrete distribution clustering (by de Carvalho et al.). Still, you do not compare your results to their method (does adding the diagonal make such a big difference?). '
>
> - Theoretically, the addition of the diagonal requires specific treatment when proving convergence and other relevant properties. It also changes fundamental properties of the space when compared to space without the diagonal (e.g., completeness, seperability), so the diagonal is vitally important in both the computation and convergence analysis. The main difference in the proofs is that any point can now be matched to the diagonal instead of just a point. Since the diagonal is of the form {(a, a): a \in \mathbb{R}} it essentially provides a ‘sink’, giving a minimal distance for each point, so it can’t just be treated like another point in the space.
>
> 'When you mention the correlation of topological complexity with generalization ability, you miss the work by Rieck et al. [2].'
>
> - We believed we had cited this and think it is an important paper, we will definitely add this citation.
>
> 'I am a fan of citing software libraries when they provide how to in their website. Hence I would recommend citing Ripser [3] (see https://github.com/Ripser/ripser for BibTeX entry) and potentially other Software you used in the main paper.'
>
> - We will add this citation too.
>
> 'You say “[…] the vectorization […] required by Lacombe et al.’s algorithm makes it unsuitable for integration into our work”. Could you elaborate on this? Can you not even choose this as a comparison partner?'
>
> - Specifically, Lacombe’s work vectorises the persistence diagram by defining a discrete grid over the space and summing the points in each area. Given that we are explicitly not mapping out of diagram space, we cannot use Lacombe’s computational speed ups. Furthermore, the clustering provided by Lacombe is hard clustering, which we compare against in Section 1.1 and in our experiments, so we essentially do compare against Lacombe’s work.
> - Note that Lacombe’s primary contribution is a computational speed up when computing barycentres of persistence diagrams. The clustering he does is not novel, but his method allows it to take place at a much increased speed. Our paper focuses on the theory of the clustering, not the computational complexity.
>
> 'What do you mean when you say other persistence-based learning strategies required prior knowledge of a ‘correct’ target topology which can’t plausibly be known? The work by Moor et al. for example, extracts persistence directly from the input, which is “trivial” to know since you can compute it. '
>
> - Apologies for being unclear here. Generally, work that adds topological loss terms requires prior knowledge of the target diagram. As you say, work on topological autoencoders compares the topology of the latent space to that of the input space, so requires no prior knowledge. We’ll edit the paper to clarify this.
>
> 'Topological Preliminaries: Why do you introduce the \v{C}ech complex when you use Vietoris-Rips for your experiments?'
>
> - The thought process was that the Cech complex is used to provide many theoretical guarantees (homotopy equivalence, for example) and is perhaps more intuitive. We will change this section to focus on the Vietoris-Rips complex.
>
> 'One feat of your approach is that you add tuples on the diagonal to make the cardinality of all compared PDs match. I wonder how susceptible your approach is to increasingly different cardinality (that has to be filled). Maybe you can conduct a small ablation study on your synthetic data set and evaluate how “good” your mean PDs are when cardinalities are increasingly different.'
>
> - This is a really interesting point, we’ll try to run this experiment before the reviewing period is over. Intuitively, we don’t think it should make a difference; if the number of significantly off-diagonal points is large, then the diagrams probably shouldn’t be clustered together, and the distance will be large. If most of the additional points are close to the diagonal, then they should probably be matched to the diagonal anyway, so it won’t matter that the added points are on the diagonal. Thanks for raising this.

---

> > ### Author Response · Authors · 2020-11-25
> > **Summary of relevant revisions**
> >
> > In our revised paper we cite the Rieck et al. paper that you suggest, as well as adding citations to the software libraries (where citations exist).  We extend our discussion of Lacombe’s work, and clarify the role of the input vs latent space in topological auto encoders. We also update our Topological Preliminaries section with the aim of making it more clear, removing mention of the Cech complex and instead focusing on the Vietoris-Rips complex. We update Figure 1 to show the Vietoris-Rips complex in support of this.

---

### Official Review · AnonReviewer2 · 2020-10-28
**A good contribution that finally enables clustering of topological descriptors**

**Rating:** 6
**Confidence:** 5

**Review:**

# Synopsis of the paper

This paper develops a novel algorithm for performing fuzzy clustering
(i.e. non-hard assignment of points to cluster centres) on persistence
diagrams, i.e. topological data descriptors. This is a highly relevant
contribution because persistence diagrams 'live' in a space that makes
metric analyses somewhat cumbersome. By contrast, the proposed method,
even though this is not strictly highlighted in the paper, can be used
as a principled way to obtain 'representatives' of a data set.

The critical algorithmic insight of the paper lies in developing a new
way to calculate Fréchet means; this makes it possible to adapt FCM to
this domain.

A set of experiments demonstrates the utility of the proposed approach.

# Summary of the review

This is a well-written paper with a highly relevant contribution for
the TDA community. I am excited to see such a clustering algorithm
finally emerge for persistence diagrams, and I envision that this paper
will be a very useful contribution to the field.

That being said, there are some issues in the current write-up that
prevent me from fully endorsing this work for now, namely:

1. Presentation: the paper will be somewhat confusing for non-experts in
   TDA. While this is to be expected to some extent, there are several
   places in which the paper could be improved to provide some more
   intuition, making it possible that even non-experts can appreciate
   the contribution.

2. The experiments appear to be somewhat preliminary. The experiment
   with synthetic data, for example, only comprises few samples that
   are clustered; a large portion of this section is spent on discussing
   an application in materials science instead (which is of course
   important, but I feel that it is hard to both appreciate the
   application domain and the algorithm at the same time). In addition,
   some details about the empirical behaviour of the method are not
   discussed.

If these two points were to be rectified in a revision of the paper,
it would help the contribution to shine more. Please see below for more
detailed comments.

# Detailed comments

- In terms of exposition, I would suggest to cite the Vietoris--Rips
  complex construction (and also refer to the complex by this
  alternative name; I think VR is much more common that just 'Rips')

- Would it not be easier to show a Vietoris--Rips filtration instead of
  the Čech one? I would suggest updating Figure 1 to account for this.
  To provide a more intuitive view to TDA, the caption of the figure
  could also be extended to describe the creation of cycles in the
  point cloud, for example. This is the first figure that readers will
  see, so any updates are well worth the effort, in my opinion.

- Footnote 2 needs some clarification: is the paper suggesting to
  compute the full VR complex (until the full simplex of $n$ vertices
  has been reached)? If not, there *will* be multiple points at
  infinity.

- Would it not be possible to sidestep the issue of infinite points
  entirely by using, say, extended persistence? It is my understanding
  that the method does not 'care' about the way the persistence diagrams
  are calculated, right? Hence, there are no structural assumptions
  being made here.

- The Fréchet mean is not necessary unique. Does this pose any problems
  for the algorithm? I would assume that the calculated clustering might
  also not be unique (or one of multiple equivalent solutions), but
  I lack the intuition here. This should be briefly discussed.

- Before equation 2, it should be 'Fréchet mean $\widehat{\mathbb{D}}$',
  i.e. the variable indicating the mean should be used here.

- How is the convergence behaviour of the algorithm? How many
  iterations does it usually take until clusters start to stabilise?

- The section on 'Computing the Fréchet mean' could be improved in terms
  of the flow. Would it be possible to provide an overview algorithm as
  well?

- In terms of the limitations of the method, it is my understanding that
  it heavily depends on the Wasserstein distance. Is this correct? I am
  asking because this distance is known to be computationally
  challenging to compute, so I was wondering while reading the paper
  whether a similar algorithm could be derived for *kernels* between
  persistence diagrams. While the experiments discuss runtime already in
  the supplements, I am not convinced about the overall scalability of
  the algorithm. (this is not a fault of the paper, I merely want the
  limitations to be discussed in more details)

- As already mentioned above, I would suggest updating the discussion of
  the lattice structure data if possible. I feel that this is an
  interesting topic, but I would rather see more experiments in the
  paper and a shortened description of the background (it could be put
  into the supplemental section).

- As an additional suggestion for improving the experiments, I would
  suggest running the synthetic test data experiments with more
  diagrams. This *can* be an excellent introductory experiment to
  showcase the capabilities of the algorithm, but at present, it falls
  slightly short of that.

- To me, the 'decision boundaries' section could be extended. This is
  a really exciting application; the ability to find representatives
  of diagrams opens up all kinds of new avenues! Is it possible to link
  this more to previous results, i.e. Ramamurthy et al.?

- As for the discussion on generalisation capabilities, I would suggest
  citing prior work (Rieck et al., 'Neural Persistence: A Complexity
  Measure for Deep Neural Networks Using Algebraic Topology'), which
  mentions relationships between topology-based measures and
  generalisation capabilities.

All in all, I am convinced that this has the potential to be a strong
addition to the TDA community!

# Style & clarity

The paper is well-written; there are a few sentences that I failed to
parse correctly, though:

- 'invariance to the basis symmetries': should this be 'basic
  symmetries' instead? Moreover, why is there an ellipsis (...) after
  'physics'? Should this be '[...]' to indicate that parts of the
  quotation were left out?

- As a matter of personal style preference, I would prefer to say 'The
  algorithm by Turner et al.' rather than "Turner et al.'s algorithm".
  The latter strikes me as somewhat confusing.

- I would suggest to use no contractions in formal writing, hence
  "cannot" instead of "can't" etc.; this is a minor point, but it since
  the remainder of the paper is written so neatly, I cannot help but
  point out ways to improve it even more.

- 'so does not' --> 'so it does not'

# Update after rebuttal & discussions

I thank the authors for their thorough rebuttal. While the technical details are acknowledged and addressed for the most part, the experimental setup could still be improved. R1 mentioned that the work by Lacombe et al. might also be applicable as a comparison partner. Investing in a more thorough scenario would strengthen the paper by a lot.

# Further update after discussions

The primary subject of our discussions concerned the experimental setup. While I still see this paper favourably, it would be strengthened by a more in-depth comparison with the work by Lacombe et al. The core of the paper would be more convincing if the utility of the fuzzy clustering could be highlighted better in a set of scenarios that are more comparable with existing TDA literature.

---

> ### Author Response · Authors · 2020-11-16
> **Initial response to reviewer**
>
> We thank Reviewer 2 for their positive review and extremely useful feedback. We're glad that you find our contribution 'novel' and 'highly relevant'. We are in the process of rectifying the two main problems highlighted in your summary, including running more experiments and providing more empirical discussion on the behaviour of the method. We also answer some of your specific questions below.
>
> 'Footnote 2 needs some clarification: is the paper suggesting to compute the full VR complex (until the full simplex of vertices has been reached)? If not, there will be multiple points at infinity.'
>
> - That is what we were suggesting. If that becomes computationally infeasible, we can continue to replace points at infinity with a suitably large value. We will update the footnote to be more clear.
>
> 'Would it not be possible to sidestep the issue of infinite points entirely by using, say, extended persistence? It is my understanding that the method does not 'care' about the way the persistence diagrams are calculated, right? Hence, there are no structural assumptions being made here.'
>
> - This is an excellent suggestion, thank you for this - we’ll update the paper to mention this!
>
> 'The Fréchet mean is not necessary unique. Does this pose any problems for the algorithm?'
>
> - This is an interesting question. I am running experiments to improve my intuition on this, and will discuss in the paper. My first intuition is that the main consequence of this will be points in the matching to compute the cluster centres may differ, which could cause problems in some (rare) cases. It may be worth slightly perturbing points in the diagrams, which intuitively I think should remove the possibility of a non-unique mean? However, I’m not sure if any theoretical guarantees of that exist.
>
> 'How is the convergence behaviour of the algorithm? How many iterations does it usually take until clusters start to stabilise?'
>
> - The convergence behaviour is generally very good. For the FCM algorithm, I would say it is generally less than 30 iterations to converge in my experience. I had no real problems running the algorithm on my laptop in any cases. Similarly, the weighted Frechet mean generally converges in <15 iterations. Note that the theoretical guarantees only guarantee convergence within a convergent subsequence, so there are cases where a fixed number of iterations is probably the best stopping criterion, however in my case it generally seemed to converge.
>
> 'The section on 'Computing the Fréchet mean' could be improved in terms of the flow. Would it be possible to provide an overview algorithm as well?'
>
> - The overview algorithm is currently in the Appendix, as we were running low on space, but given the additional page of space for the revisions we’ll look at moving it into the main body of the paper.
>
> 'In terms of the limitations of the method, it is my understanding that it heavily depends on the Wasserstein distance. Is this correct? I am asking because this distance is known to be computationally challenging to compute, so I was wondering while reading the paper whether a similar algorithm could be derived for kernels between persistence diagrams.'
>
> - You are correct that the computation of the Wasserstein distance (or, more specifically, the optimal matching between diagrams) is the main bottleneck for our algorithm. Using a kernel between diagrams for clustering is a very interesting idea. The algorithm itself should be easily adapted to using a kernel, and since the Wasserstein distance has weaker theoretical guarantees than a positive-definite kernel, the convergence proofs should be easily extended to kernels. This is a really good idea for an extension, and would certainly speed up our algorithm, making it more accessible for computationally more difficult problems.
>
> 'As an additional suggestion for improving the experiments, I would suggest running the synthetic test data experiments with more diagrams.'
>
> - We will do that before the end of the reponse window.
>
> To me, the 'decision boundaries' section could be extended. This is a really exciting application; the ability to find representatives of diagrams opens up all kinds of new avenues! Is it possible to link this more to previous results, i.e. Ramamurthy et al.?
>
> - I agree that is an exciting direction, we will extend the discussion to include more relevant previous results.
>
> As for the discussion on generalisation capabilities, I would suggest citing prior work (Rieck et al., 'Neural Persistence: A Complexity Measure for Deep Neural Networks Using Algebraic Topology')'
>
> - I actually thought I’d cited this paper, I’ll definitely add it in!
>
> ''invariance to the basis symmetries': should this be 'basic symmetries' instead? Moreover, why is there an ellipsis (...) after 'physics'?'
>
> - The quote does say basis symmetries, I think in the linear algebra sense. It should be […], apologies for that.
>
> Thanks again for such a comprehensive review, with many useful suggestions.

---

> > ### Comment · AnonReviewer2 · 2020-11-18
> > **Thanks for the clarification**
> >
> > Thanks for the clarification! Would it be possible to add some supplementary plots about the empirical convergence behaviour? This would be a nice way to further support the algorithm. Moreover, are there cases in which the algorithm failed to convergence? I am asking this as a potential user of the method.

---

> > > ### Author Response · Authors · 2020-11-25
> > > **Empirical convergence plots and summary of relevant revisions**
> > >
> > > Please see the new Figure 3, where we’ve added the requested supplementary plots analysing the empirical performance behaviour. In all experiments ran, our algorithm did not fail to converge (convergence defined by the cost function not fluctuating by more than +-0.5%). We found in practice that stopping the algorithm after a fixed number of iterations (which is commonly done for Euclidean FCM) is the easiest way to use the algorithm, and in doing so we found we reached cluster centres and membership values that were useful to our applications.
> > >
> > > To highlight changes made to our paper on the basis of your review, we’ve updated Footnote 2 to be more clear and mention extended persistence as a solution to the points at infinity. We update the topological preliminaries section to hopefully provide a more intuitive introduction to topology, switching to only introducing Vietoris-Rips complexes (including in Figure 1, which we remade). We have added a section to the Synthetic Data experiments to discuss the empirical behaviour of the algorithm, as well as providing plots with average iterations for convergence with varying number of diagrams and points per diagram. We did not end up with space to move the overview algorithm into the main body of the paper, but we’ve highlighted that it is available in the appendices. The synthetic test data was ran with more diagrams for the empirical analysis. We include the reference to Rieck’s paper in the discussion of the decision boundaries section.

---

### Official Review · AnonReviewer4 · 2020-10-29
**This is a incremental work to cluster persistence diagrams**

**Rating:** 3
**Confidence:** 5

**Review:**

This paper aims to apply Fuzzy c-Means (FCM) clustering to persistence diagrams and prove convergent subsequence of iterates tends to a local minimum or saddle point. The motivation of the work is direct and clear. This paper addresses the problem of persistence diagram clustering via using  weighted Frechet mean .

This is a incremental work though replacing Euclidean distance in FCM by Wasserstein distance. The contributions in this work are not quite promising. For instance, the weighted Frechet mean in Sec3.2 is the well-known Wasserstein Barycenter whose behavior is well studied in optimal transport works. Thus Theorem2 can be not be considered as the contribution of this work.

Another drawback is that experiment in this work is very weak. There are only three dataset tested in this work. It's not quite convincing. It would be promising if the proposed work valid in other shape datasets such as SHREC2010 or SHREC2014. These datasets were frequently used for testing algorithms of topological data analysis .

I cannot recognize the merits of this work compared with previous papers, such as: Large Scale computation of Means and Clusters for Persistence Diagrams using Optimal Transport.

---

> ### Author Response · Authors · 2020-11-16
> **Initial reviewer response**
>
> Hi, thank you for your review. We respond to your points directly below. In particular, we disagree with your characterisation of our work as simply replacing the Euclidean distance with the Wasserstein distance. We go into more details below, but the space of diagrams endowed with the Wasserstein distance is a space with far weaker properties than Euclidean space, so requires significant additional theoretical work (which we do) to compute the cluster centres and prove convergence properties for the algorithm.
>
> 'This is a incremental work though replacing Euclidean distance in FCM by Wasserstein distance.'
>
> - When computing the membership values, we do simply replace the Euclidean distance. However, the computation of the cluster centres is different to that of Euclidean FCM. Perhaps more importantly than giving the algorithm, however, we provide the theoretical analysis to give convergence guarantees for FCM in the space of persistence diagrams. This convergence analysis has not been done before, and goes beyond simply updating the Euclidean distance with the Wasserstein distance.
>
> 'The weighted Frechet mean in Sec3.2 is the well-known Wasserstein Barycenter whose behavior is well studied in optimal transport works. Thus Theorem2 can be not be considered as the contribution of this work.'
>
> - Although the weighted Frechet mean of persistence diagrams is similar to the Euclidean Wasserstein Barycentre, it is distinct. In particular, the presence of the diagonal {(a,a) : a \in \mathbb{R}} with infinite multiplicity makes both the computation of the Barycentre and the convergence analysis different to the Euclidean case. Therefore we believe that Theorem 2 stands as a contribution of this paper.
>
> 'Another drawback is that experiment in this work is very weak. There are only three dataset tested in this work. It's not quite convincing. It would be promising if the proposed work valid in other shape datasets such as SHREC2010 or SHREC2014.'
>
> - Thank you for your feedback on the experimental section. Due to it and other reviewers feedback we are working on additional experiments, including using your recommended shape datasets SCHREC2010 and SCHREC 2014. We would point out that we conduct experiments on five datasets, namely: the synthetic data, BCC/FCC data from the Materials Project, carbon allotropes from the Samara Carbon Allotropes dataset, MNIST, Fashion-MNIST, and Kuzushiji MNIST.

---

> > ### Author Response · Authors · 2020-11-25
> > **Summary of relevant revisions**
> >
> > We updated our paper to clarify that the space of diagrams has weaker theoretical properties than Euclidean space, so our paper requires significant additional work in comparison to the Euclidean case, both to compute cluster centres and to give convergence guarantees.
> >
> > We were unable to gain access to the SHREC datasets within a reasonable timeframe, as the copies we initially found on the internet were all password protected, so were unfortunately unable to run experiments on the SHREC dataset. However, we did run additional experiments on synthetic data to evaluate the convergence properties of our algorithm in practice, which we hope you’ll find interesting.

---

### Official Review · AnonReviewer1 · 2020-10-30
**Not Quite Exciting**

**Rating:** 4
**Confidence:** 4

**Review:**

The paper proposes a new clustering algorithm for persistence diagrams. They use fuzzy c means clustering. The motivation for fuzzy clustering is that it allows each datum (persistence diagram) to have weighted (soft) membership in different clusters. The partial membership value is the ratio of the distance to that cluster center to sum of all memberships to other clusters. Empirical results on synthetic and real data shows that clustering of persistence diagrams outperforms others that depends on geometry. A convergence theorem is provided, based on previous fuzzy k-means convergence proof.

This paper is well written. The clustering of persistence diagrams is an important practical problem and the proposed algorithm does seem to work.

However, overall the benefit of the new soft-membership clustering is not clear. The authors claim that (Lacombe et al. NeurIPS 2018) is hard clustering. But indeed (Lacombe et al. 18) relaxed the diagram to a continuous function. So it is unclear why it cannot have some of the benefits of the proposed method. Also the following paper needs to be discussed.

Persistence Bag-of-Words for Topological Data Analysis - IJCAI'19

Empirically, some important tables should be included in the main paper. It is difficult to find the results in the supplemental material. The comparison to hard membership is hard to find. Also maybe Lacombe et al should be compared with?

Furthermore, the real world data experiments is very specially chosen to show that when geometric transformation is involved, topology-based method is better than geometry-dependent baselines. But this is kind of obvious. And there are definitely better adaptations of these baselines to be more robust to these geometric transformations (e.g., using ICP algorithm to compute distance for WBC). I suggest the authors look for better dataset (ideally datasets used by supervised methods) and show that PH based clustering algorithm is useful in these datasets.

I think the bottomline to me is that the key ideas are really not quite surprising.

------------

I have read the authors' response. I do not think the answers addressed my concerns.

On the positive side, I do think the paper is written well and the idea is clear. It is true that a soft-clustering of PDs on the true Wasserstein distance has not been done. The paper also provided proof of the convergence of the algorithm.

My main concerns are the following:

First and most importantly, the method is not particularly surprising. It basically combines the most classic soft-clustering algorithm and the Frechet mean computation algorithm of Turner et al. Theoretically, the proof is extending the convergence result of the soft-clustering algorithm to the Wasserstein distance of PDs. I would not say the proof is trivial. But it is not that surprising, as we already knew that the Frechet mean of PDs is computable. I would be much more excited if the theoretical result is about the optimality rather than just convergence.

Empirically, the paper did not provide comparison with (Latombe et al. 18). Just saying that they did not do it exactly in the PD space is not good enough. A lot of practically powerful methods (persistence image, various kernels, etc) are approximations/relaxations outside the PD space. These approximations/relaxations can bring computational advantage, and sometimes better learning efficiency. Therefore, we need to know how this method is compared with (Latombe et al. 18) in efficiency and clustering performance. (Latombe et al. 18) can naturally have both hard- and soft-clustering versions. A thorough comparison with the different versions can show how important it is to stick with the PD space rather than the relaxation. My guess is that in practice sticking with PD space is not that important, or maybe even worse due to bad local optima of the Frechet mean. But I would be very happy to be proven wrong.

Another issue is the limited experiments. The material data does seem to be a good fit. But the authors could use some of the classic topology-friendly data (shape, dynamic data, graph) from existing supervised methods. Any labeled binary/multiclass data can be used to evaluate clustering. An even more ambitious goal is to prove the usefulness of clustering in the supervised task. For example, the authors can show that a bag-of-words approach (using the clustering result) can improve classification performance.

Overall, I feel that the methodology is not very exciting to me, and the experiments are insufficient. If the main argument is the algorithm computes on the PD space and the proof of convergence, this paper may better fit a theoretical conference.

---

> ### Author Response · Authors · 2020-11-16
> **Initial reviewer response**
>
> Hi, thank you for your review. We appreciate that you find our paper 'well written' and believe that 'the clustering of persistence diagrams is an important practical problem and the proposed algorithm does seem to work'. We've addressed your concerns directly below. In particular, we maintain that Lacombe's paper only performs hard clustering.
>
> 'The authors claim that (Lacombe et al. NeurIPS 2018) is hard clustering. But indeed (Lacombe et al. 18) relaxed the diagram to a continuous function. So it is unclear why it cannot have some of the benefits of the proposed method.'
>
> - We're unsure which relaxation you’re referring to in Lacombe’s paper. They (i) embed the diagram in Euclidean space by summing the points in a discrete grid, and (ii) relax the barycentre computation problem by applying entropic smoothing. However, the embedding in (i) takes us out of diagram space, which we are explicitly trying to avoid, and is not related to the labels, and (ii) is to smooth the objective function, not to relax the labels. We believe we are correct in our statement that Lacombe only implements hard clustering.
>
> 'Also the following paper needs to be discussed: Persistence Bag-of-Words for Topological Data Analysis - IJCAI'19.'
>
> - This paper introduces a vectorisation technique for persistence diagrams. We are explicitly avoiding embedding persistence diagrams, but do summarise the major techniques for embedding in the background section - we will add this paper to that list.
>
> 'Empirically, some important tables should be included in the main paper. It is difficult to find the results in the supplemental material. The comparison to hard membership is hard to find.'
>
> - We’re happy to bring important tables into the main paper. Unfortunately the page limit meant it was hard to know which aspects of the work to highlight, so we appreciate the feedback on which results to include.
>
> 'Furthermore, the real world data experiments is very specially chosen to show that when geometric transformation is involved, topology-based method is better than geometry-dependent baselines.'
>
> - The materials dataset was chosen as an example of a dataset that is far better suited to topological techniques than traditional geometric techniques. If geometric techniques were better, we believe it would make more sense to just use them! The purpose of our paper is to add a traditionally very powerful unsupervised learning algorithm to TDA’s repertoire - therefore we demonstrate it on datasets better suited to TDA.
> - In addition, we believe that its far from clear that topology would be able to match pre-trained decision boundaries to unseen tasks using our method, so this data is not specifically chosen for TDA, but hopefully demonstrates the power of TDA for model selection.

---

> > ### Author Response · Authors · 2020-11-25
> > **Summary of relevant revisions**
> >
> > In our revised version we’ve added your recommended paper ‘Persistence Bag-of-Words for Topological Data Analysis’ to our discussion on embedding techniques for persistence diagrams.

---

### Decision · Program_Chairs · 2021-01-07
**Final Decision**

**Decision:**

Reject

**Comment:**

This paper is overall clearly written, and the proposed approach of performing clustering on the space of persistence diagrams can be a significant contribution.

However, during the discussion, the reviewers share the concern about insufficient empirical evaluation. In particular, datasets are limited and (Lacombe et al. 2018) is not included as a comparison partner, although the authors had a chance to include it in the author response phase. Since this point is crucial, I will reject the paper.

Addressing these points will largely improve the paper, and also reviewers put a great effort to give detailed reviews for the paper. So I hope the authors take the reviews into consideration for further revision of the paper.